# TGF-β Induced by Allergic Lung Inflammation Enhances Os-Teosarcoma Lung Metastasis in a Mouse Comorbidity Model

**DOI:** 10.3390/ijms26115073

**Published:** 2025-05-24

**Authors:** Marco J. Sanchez-Rojas, Belen Tirado-Rodriguez, Gabriela Antonio-Andres, Giovanny Soca-Chafre, Daniel D. Hernandez-Cueto, Cesar O. Martinez-Calderon, Mayra Montecillo-Aguado, Juan C. Hernandez-Guerrero, Marco A. Duran-Padilla, Rogelio Hernandez-Pando, Sara Huerta-Yepez

**Affiliations:** 1Unidad de Investigacion en Enfermedades Oncologicas, Hospital Infantil de Mexico, Federico Gomez, Mexico City 06720, Mexico; mjsrojas@hotmail.com (M.J.S.-R.); abtirado81@gmail.com (B.T.-R.); gabya_24@yahoo.com.mx (G.A.-A.); g.s.ch@outlook.com (G.S.-C.); danteligueri@gmail.com (D.D.H.-C.); cesarm1926@gmail.com (C.O.M.-C.); mayramontecillo@gmail.com (M.M.-A.); 2Programa de Doctorado en Ciencias Biologicas, Universidad Nacional Autonoma de, Mexico (UNAM), Mexico City 04510, Mexico; 3Laboratorio de Biomedicina Traslacional, División Ciencias de la Salud, Universidad de Guanajuato, Guanajuato 36000, Mexico; 4Laboratorio de Inmunología, Facultad de Odontología, Mexico City 04510, Mexico; medicinabucaljcc@yahoo.com.mx; 5Unidad de Patología Quirúrgica, Hospital General de México Dr. Eduardo Liceaga, Mexico City 04510, Mexico; patologiaduran@hotmail.com; 6Seccion de Patología Experimental, Departamento de Patología, Instituto Nacional de Ciencias Médicas y Nutrición Salvador Zubirán (INCMNSZ), Mexico City 14080, Mexico; rogelio.hernandezp@incmnsz.com

**Keywords:** allergic airway inflammation, osteosarcoma, lung metastasis, TGF-β, PCNA, differential gene, expression, overall survival, prognosis, AllergoOncology

## Abstract

TGF-β is a central mediator of pulmonary allergic inflammation recently associated with lung metastasis of osteosarcoma. Given the controversial links between cancer and allergic diseases, this study aimed to evaluate the effects of allergic airway inflammation—particularly TGF-β—on osteosarcoma lung metastasis using a comorbidity mouse model. Osteosarcoma cells were implanted in BALB/c mice with induced allergic airway inflammation. Lung metastasis was quantified, while PCNA/TGF-β expression was assessed by immunohistochemistry and digital pathology. Bioinformatic analyses of patient datasets compared TGF-β and PCNA expression in metastatic vs. normal tissues, and their association with survival. Mice with allergic inflammation showed increased lung metastases associated with TGF-β production. In patient samples, both TGF-β and PCNA were upregulated in metastatic tissues and correlated with poor overall survival. PCNA was also linked to genes involved in cell proliferation, DNA replication, and repair. Our results show an association between allergic airway inflammation and extensive lung metastasis of osteosarcoma in a comorbidity mouse model with elevated expression of TGF-β and PCNA.

## 1. Introduction

Osteosarcoma is the most common malignant bone tumor predominantly affecting children and adolescents. It arises from primitive bone-forming mesenchymal cells, characterized by osteoid tissue production. Global incidence of osteosarcoma is approximately three to four cases per million annually, with peak incidence in the second decade of life, coinciding with periods of rapid bone growth. Despite advances in multi-agent chemotherapy and surgical techniques, osteosarcoma prognosis remains poor, particularly in metastatic disease. Lung metastasis is most frequent, occurring in approximately 40–50% of patients [1,2,3]. The 5-year survival rate for patients with localized osteosarcoma reaches 60–70%, but falls to 20–30% for those with metastatic disease. Metastasis challenges osteosarcoma management given its refractory and aggressive nature. Current therapies often fail to effectively prevent or control tumor spread. Understanding its molecular and cellular mechanisms is critical to target metastatic behavior and improve long-term prognosis [3,4].

The transforming growth factor-beta (TGF-β) signaling pathway is among the key molecular factors implicated in osteosarcoma metastasis. TGF-β is a multifunctional cytokine controlling diverse cellular processes, including cell proliferation, differentiation, as well as immune response [5]. In cancer, TGF-β plays dual roles, acting as either a tumor suppressor or promoter in early and advanced stages, respectively [6,7]. Recent studies highlight its pivotal role in facilitating lung metastasis by osteosarcoma, promoting epithelial-to-mesenchymal transition (EMT), affecting cell motility, and the tumor microenvironment (TME). TGF-β overexpression has been found in osteosarcoma TME, associated with increased metastasis and poor clinical outcomes. Additionally, TGF-β induces immunosuppression in metastatic sites, allowing osteosarcoma cells to evade immune surveillance, establishing secondary lung tumors. Targeting the TGF-β pathway in osteosarcoma is therefore a promising therapeutic strategy to prevent lung metastasis and improve patient survival [8,9].

On the other hand, allergic lung inflammation, such as asthma, is characterized by airway hyperresponsiveness, mucus overproduction, and inflammatory infiltrate, becoming a hallmark of chronic respiratory diseases. In this context, TGF-β is increasingly recognized as a central mediator of chronic inflammation and airway remodeling [10,11]. During an allergic response, there is an elevated production of TGF-β by epithelial cells, macrophages, and activated immune cells in the lungs. TGF-β promotes differentiation of regulatory (Tregs) and helper 2 (Th2) T cells, which are key drivers of allergic inflammation. This cytokine also activates fibroblasts, leading to pulmonary fibrosis and airway remodeling, impairing lung function in chronic allergic conditions such as asthma and bronchitis. Moreover, TGF-β modulates the extracellular matrix (ECM), promoting airway smooth muscle (ASM) proliferation, while exacerbating airway narrowing and resistance. Its role in chronic inflammation, tissue damage, and fibrosis makes TGF-β a crucial target for therapeutic intervention in allergic lung diseases [11].

In recent years, AllergoOncology has developed as a field studying relationships between allergic diseases and cancer. Some studies show allergy may be protective against certain cancer types, while others indicate allergic conditions could stimulate tumor development, such as the large cohort study conducted by Karim et al. (2019), which demonstrated that patients with allergic rhinitis had a lower risk of developing several malignancies, including breast and prostate cancer [12]. Conversely, research by Woo et al. (2021) indicated that certain allergic conditions, such as asthma, may increase lung cancer risk. These findings highlight the complex interplays between allergic immune response and tumor biology, warranting further investigation into their influence in cancer development and progression [13,14,15]. In this study, we aimed to investigate the effects of allergic lung inflammation and TGF-β on lung metastasis of osteosarcoma. Using a murine comorbidity model, we explored whether elevated TGF-β levels typically produced during allergic inflammation are associated with increased metastatic potential of osteosarcoma cells in the lungs.

## 2. Results

### 2.1. Allergic Airway Inflammation Promotes Lung Metastasis in Mice

To evaluate whether allergic inflammation affects tumor progression, we used a mouse model of allergic airway inflammation induced by OVA, followed by the administration of osteosarcoma (OS) cell line K7M2 in the cortical diaphysis of the tibia. After 32 days, the animals were euthanized, and their lungs were analyzed to determine the number and size of metastatic nodules (Figure 1A). Mice with allergic inflammation induced by OVA show peribronchial and perivascular extensive chronic inflammation with hyperplastic epithelium covered by a thick mucus layer, while control mice treated with saline solution (SS) exhibited occasional lymphocytes in the vascular and airway walls (Figure 1B). Next, we implanted OS cells via intratibial injection that produced infiltrating osteosarcoma predominantly constituted by fusiform cells and osteoid production with occasional small or medium-size pulmonary metastatic nodules (Figure 1C). Meanwhile, asthmatic mice showed similar bone tumors with numerous metastatic nodules of variable size randomly distributed in the lungs (Figure 1C). This difference was confirmed by digital pathology, revealing a significantly larger lung area with metastatic nodules (*p* = 0.030) in OVA + OS mice, as well as a higher mitotic index (*p* = 0.001) (Figure 1C).

Histopathological examination (Figure 1C) shows highly aggressive and invasive patterns of tumor cells in the lungs of the OVA + OS group (Figure 1(Cb)) compared to the SS + OS group (Figure 1(Ca)). Multiple tumor nodules predominantly located in peribronchial areas were observed in the OVA + OS group; these were solid tumors limited by thin septa of connective tissue. Neoplastic cells were large, ovoid, and round, with abundant eosinophilic cytoplasm, large hyperchromatic nuclei with prominent nucleoli, and numerous mitotic figures. No significant inflammatory intratumoral response was observed, while in several bronchi their luminal spaces contained numerous neutrophils, histiocytes, lymphocytes, plasma cells, and occasional cellular debris (Figure 1C). Additionally, smaller nodules of tumor cells were observed, accompanied in the periphery by xanthomatous inflammation mostly composed of foamy macrophages within alveolar spaces. This suggests that inflammation in SS + OS mice derives from tumor compression of pulmonary bronchioles, leading to a mixed inflammatory pattern of pneumonia characterized by intra-alveolar polymorphonuclear cells, macrophages, and lymphocytes. Tumor invasion into blood and lymphatic vessels was also seen.

In contrast, in SS + OS mice (Figure 1(Cc)), some smaller or mid-sized metastatic nodules were observed with similar sarcomatous histology as bone tumors, predominantly located in subpleural and intraparenchymal regions with peribronchial and perivascular distribution. Inflammatory infiltrates in OVA + OS mice (Figure 1(Cd)) were primarily associated with tumor cells but not with surrounding parenchyma. These results show that allergic inflammation is significantly associated with lung metastasis possibly promoting tumor aggressiveness and invasion.

### 2.2. Pulmonary Allergic Inflammation Upregulates TGF-β Expression and Lung Metastasis of Osteosarcoma

To evaluate the immune response related to lung metastasis in our model, the serum concentration of several cytokines was determined using a CBA (cytometric bead array). Interestingly, TGF-β displayed the highest serum level among other cytokines in OVA compared to SS mice (Appendix A). Next, we contrasted TGF-β expression in lung tissue by IHC. SS + OS mice showed some neoplastic positive cells, while OVA + OS mice showed strong TGF-β immunostaining in the cytoplasm of bronchial epithelial cells and in numerous cells of the perivascular and peribronchial inflammatory infiltrate. Many neoplastic cells also showed strong immunostaining as well as alveolar and interstitial macrophages located around of metastatic nodules (Figure 2A). Thus, pulmonary allergic inflammation and osteogenic sarcoma pulmonary metastasis are associated with higher TGF-β expression. This observation was confirmed by digital pathology showing a statistically significant difference between these groups (*p* = 0.035) (Figure 2B). These observations indicate that allergic inflammation leads to TGF-β overexpression, which may play a role in promoting lung metastasis of osteogenic sarcoma.

### 2.3. Pulmonary Allergic Inflammation Increases Neoplastic Cell Proliferation in Lung Metastasis of Osteosarcoma

In advanced stages, TGF-β drives tumor cell proliferation, often reflected by high PCNA expression [16]. To investigate whether this mechanism could underlie enhanced lung metastasis observed in our murine model, we analyzed cell proliferation by IHC staining for PCNA (Figure 3A). Interestingly, many neoplastic cells from SS + OS mice were PCNA-positive, but they were more frequent in the OVA + OS group. Quantification by digital pathology confirmed this observation showing statistically significant differences between these groups (*p* = 0.035) (Figure 3B). These findings support how in the context of pulmonary allergy, increased lung metastasis is associated with enhanced proliferation of osteosarcoma cells.

### 2.4. Database Analysis

Analysis of the database revealed that patients with lung metastasis had significantly higher TGF-β and PCNA expression compared to normal tissue (Figure 4A), associated with shorter overall survival (Figure 4B). Hence, overexpression of these markers may account for a more aggressive tumor phenotype and worse clinical outcomes. Interestingly, higher PCNA associated with cell proliferation was present in pre-chemotherapy biopsies in high-grade metastatic (META) vs. non-metastatic (NON-META) osteosarcoma (Figure 5A). This correlated with poorer survival in patients who developed metastasis within 5 years vs. those who did not (*p* = 0.014) with increased risk of death by 300% (HR = 3.063). (Figure 5B). Furthermore, expression profiles of several proteins interacting with PCNA in cell proliferation-related processes (Figure 6) showed upregulation in metastatic (META) vs. non-metastatic (NON-META) osteosarcoma. These include PCNA-associated recombination inhibitor (PARI), DNA polymerases PolH and PolD1, MutS homologues 3 and 6 (MSH3, MSH6), exonuclease 1 (Exo1), DNA ligase 1 (Lig1), F-box DNA helicase 1 (FBH1) and zinc finger RANBP2-type containing 3 (ZRANB3). These findings indicate how TGF-β and PCNA may represent potential biomarkers to improve prognosis and treatment in osteosarcoma.

## 3. Discussion

There are conflicting observations regarding the relationships between allergic diseases and cancer. While some studies suggest protective roles for allergic inflammation in certain cancer types [13,17,18], others report pro-tumorigenic effects [11,19]. However, studies on co-existing pulmonary allergic inflammation and osteosarcoma in mice are lacking. Addressing this gap was the primary objective of the present study.

Osteosarcoma is a highly aggressive bone sarcoma primarily affecting children, adolescents, and young adults. At the time of diagnosis, approximately 20% of patients present pulmonary metastases, and 40% develop pulmonary metastases later during the disease course. Unfortunately, survival outcomes for patients with recurrent osteosarcoma and pulmonary metastases have remained largely unchanged over the past decades, despite therapeutic advancements [20].

TGF-β is a critical immunosuppressive molecule within osteosarcoma TME. High TGF-β production inhibits cytotoxic T and natural killer (NK) cells, both of which play essential roles in antitumor immunity. Elevated TGF-β levels are strongly associated with high-grade osteosarcoma and pulmonary metastases, highlighting its pivotal role in promoting immune evasion and tumor progression [21]. Disrupting TGF-β signaling in mice with urethane-induced lung cancer significantly stimulated tumor progression [22]. This finding emphasizes TGF-β activity for tumor suppression in chemically induced lung cancer. High TGF-β has been observed in allergic lung inflammation, such as asthma. Thus, in the present study, we used a well-established murine allergic airway inflammation model [23] to analyze osteosarcoma progression in relation to TGF-β expression. We compared metastatic areas in the lungs of allergic vs. non-allergic mice. Our data showed that pulmonary allergy is associated with osteosarcoma metastasis in this organ. Our study is the first to demonstrate the effects of pulmonary allergic inflammation, characteristic of asthma, on osteosarcoma progression.

Our results showed high TGF-β in serum and tumor tissue in the OVA + OS group compared to SS + OS mice (Appendix A). TGF-β overexpression has been reported in allergy, correlated with disease severity [22,24,25], which was in agreement with high TGF-β levels found in our allergic mice. In a previous study, we found an inverse relationship between allergic airway inflammation and tumor progression in a model of breast cancer in mice: TGF-β overexpression during allergic airway inflammation induced apoptosis and inhibited cell proliferation, thereby impairing tumor progression [16]. These findings contrast with our observations in the current study, where allergy coexists with osteosarcoma progression. TGF-β signaling displays a differential behavior in different cancer types [26]. In most epithelial and hematopoietic carcinomas, TGF-β signaling acts as a tumor suppressor during early stages, as observed in our previous allergy and breast cancer model, where we evaluated the TGF-β effect on primary tumors. Contrarily, as tumors progress, TGF-β signaling often promotes metastasis [27]. Consistent with our current findings on pulmonary metastases, in advanced cancers TGF-β stimulates tumor cell proliferation, survival, and metastasis by enhancing cellular processes such as epithelial-to-mesenchymal transition (EMT), facilitating cancer cell invasion and spread.

Controversially, some studies have shown TGFβ inhibition of cell cycle progression in G1 phase through cyclin-dependent kinase inhibitors and p21, downregulating MYC expression, a crucial proto-oncogene for cell proliferation [28,29]. In contrast, aberrant TGFβ upregulation in TME is implicated in angiogenesis and bone remodeling, driving osteosarcoma progression and metastasis [30]. For this reason, we were particularly interested in evaluating whether PCNA expression, as a cell proliferation marker, was affected in our experiment. Interestingly, we observed higher PCNA levels in OVA + OS vs. SS + OS mice. TGF-β is known to activate cell proliferation through multiple signaling pathways [31], including ERK signaling, where TGF-β stimulates TβRI receptors, which in turn activate ERK signaling via the SHCA-GRB2-SOS complex, initiating a MAPK cascade involving RAS, RAF, MEK, and ERK. Activated ERK regulates cell survival, proliferation, and migration, contributing to tumor progression. Another mechanism involves TGF-β activation of p38 and JNK through TRAF/TAK1 signaling, independent of receptor kinase activity. These MAPKs regulate cellular stress responses stimulating tumor proliferation and development. TGF-β also activates the PI3K/AKT signaling pathway leading to AKT phosphorylation. AKT promotes cell survival, growth, and proliferation by targeting downstream effectors such as MTOR and FOXO transcription factors. While these mechanisms are well-established in the context of TGF-β signaling, further studies are required to clarify which specific pathway(s) are implicated in enhanced cell proliferation observed in our experimental model [31,32,33].

TGF-β indeed participates in tumor cell proliferation, particularly during cancer progression. This observation aligns with results from our bioinformatic analysis, where lung metastasis exhibited significantly higher TGF-β and PCNA expression vs. normal tissue, associated with reduced patient survival. Clinically, higher PCNA levels were detected in metastatic vs. non-metastatic osteosarcoma, representing a poor prognosis for these patients. Additionally, multiple enzymes directly interacting with PCNA in DNA replication and repair were up-regulated, indicating activation of these pathways [34,35]. These findings suggest that high expression of these proteins may serve as biomarkers of more aggressive tumor phenotypes and clinical outcomes.

As previously mentioned, TGF-β is now widely recognized as a tumor suppressor during early stages but becomes an oncogene in advanced settings [30]. In addition, TGF-β also exhibits predominant tumorigenic behavior in osteosarcoma, enhancing cancer cell migration, invasion, and lung metastasis [36]. Based on these facts, we hypothesized and confirmed how TGF-β overexpression, driven by a concomitant condition such as allergic disease, stimulates pulmonary metastasis in osteosarcoma. Furthermore, it was critical to recreate this process within a model with coexisting rather than independent diseases. The interplay between TGF-β, allergic inflammation, and tumor progression in this context may reveal unique interactions not apparent in individual diseases. By using this integrated model, we provide a more comprehensive understanding of TGF-β role in pulmonary metastasis. This approach better represents the complex interactions between tumor biology and immune modulation in a dual-disease setting, offering new insights into potential therapeutic strategies for managing osteosarcoma with concomitant allergic conditions. However, we did not evaluate other metastasis-related factors, such as systemic inflammation and immunosuppression.

In multiple malignancies, TGF-β signaling displays oncogenic effects by promoting metastasis, angiogenesis, and immune evasion. For example, in breast cancer, transcription factor SOX4 activates TGF-β signaling, promoting EMT, cancer progression, and eventually metastasis [37]. In prostate cancer, TGF-β stimulates cellular migration, triggering cytoskeletal rearrangement and cell cycle activation through several proteins, including the SMAD family, Cdc42, and Rho A. In pancreatic ductal adenocarcinoma, blocking TGF-β-mediated fibroblasts activation prevents cell proliferation, tumor development and metastatic spread [38]. In colon cancer, dual inhibition of TGF-β receptors I and II, significantly impairs cell migration and invasion ultimately reducing liver metastasis and improving survival in mouse models [39]. Therefore, drugs targeting TGF-β and its receptors are being developed for multiple malignant tumors in the metastatic stage, including gastric and colorectal cancer, as well as progressive glioma, among many others [40].

## 4. Materials and Methods

### 4.1. Reagents, Antibodies, and Cell Line

Ovalbumin (OVA) grade V was obtained from Sigma-Aldrich (St. Louis, MO, USA). Aluminum hydroxide (alum) was acquired from Pierce Biotechnology (Rockford, IL, USA). Antibodies anti-TGF-β and anti-Proliferating cell nuclear antigen (PCNA) were purchased from Santa Cruz Biotechnology (Santa Cruz, CA, USA). Normal rabbit IgG was used as an isotype control (IC).

Osteosarcoma K7M2 cell line was supplied by the American Type Culture Collection (ATCC), catalogue number CRL-2188. K7M2 cells were cultured in Dulbecco’s Modified Eagle Medium (DMEM) supplemented with 10% fetal bovine serum (FBS) and 1% penicillin–streptomycin. Cells were maintained at 37 °C in a humidified atmosphere containing 5% CO_2_.

### 4.2. Animals and Ethical Statements

BALB/c mice, male (specific pathogen-free, SPF), aged 6 weeks (16–18 g), were housed in pathogen-free conditions. The animals were housed in the same unit under sterile conditions in a rack with airflow of 20 changes per hour, relative humidity of 45 to 65% at a temperature between 18 and 22 °C. Animals were obtained from the animal facility at the Hospital Infantil de México Federico Gómez (HIMFG). Mice received sterilized food and water ad libitum and were maintained in a ventilated rack under sterile conditions with temperature control and 12-h light/dark cycles. All animal handling and care were conducted according to the Mexican Official Standard NOM-062-ZOO-1999 [41], which outlines technical specifications for the production, care, and use of laboratory animals; and approved by the Internal Committee for the Care and Use of Laboratory Animals of the Hospital Infantil de Mexico (HIM/2017/014). Sample size calculation was based on power analysis and a previous publication from our group under similar experimental conditions, i.e., measuring TGF-β by IHC in mice with OVA-induced allergic lung inflammation [16]. Effect size was measured for a two-sided independent t test, whereas confidence level and power were set at 95% and 80%, respectively, using R software version 4.4.1. The results showed that the sample size adjusted for a 20% death rate was 4 mice per group. Further contrast with the resource equation for animal studies [42] led us to select 6 as an upper limit; therefore, we used 2 groups with 4–6 animals each.

### 4.3. Allergic Airway Inflammation Model

BALB/c mice are commonly used in asthma models due to their strong Th2-skewed immune response, which closely mimics the pathophysiology of human allergic asthma. They exhibit robust airway inflammation, eosinophilia, and elevated IgE levels upon allergen exposure. This makes them ideal for studying allergic airway diseases [43,44,45]. We previously published several studies utilizing this well-established allergic lung inflammation model using BALB/c mice [16,46,47,48,49]. Briefly, male Balb/c mice were selected based on predisposition to develop a Th2 response to OVA (Grade V, Sigma-Aldrich, Darmstadt, Germany). Six-week-old animals were split into a control group receiving saline solution (SS) and a group with allergic lung inflammation induced by OVA. This was accomplished by two allergen sensitizations via intraperitoneal (i.p.) injections. Subsequently, an allergic challenge was performed to induce and maintain Th2 responses typical of asthmatic processes in mice. Mice were either inoculated with the OVA allergen directly via the intratracheal route or not. The allergic lung inflammation was observed 10 days after the last OVA challenge.

### 4.4. Experimental Design for the Osteosarcoma Model (K7M2 Cells)

After establishing allergic responses in mice, K7M2 cells were surgically implanted as previously described [50,51]. Briefly, mice were placed in an anesthesia induction chamber, and treated with isoflurane plus 100% oxygen at 2.5 L/min until loss of righting reflex. The animals were transferred to a heating pad and maintained under general anesthesia using a nasal cone at a flow rate of 2.5 L/min. The hind leg was cleaned with betadine followed by 70% ethanol. A small hole was created in the center of the tibia using a 17-gauge needle, angled approximately 45° in the sagittal plane and advanced until it trespassed the bone cortex, where 1 × 10^6^ K7M2 osteosarcoma cells suspended in 50 µL PBS were directly inoculated. After 54 days, the mice were euthanized, and peripheral blood and lungs were collected. Serum was stored at −80 °C, and lungs were perfused intratracheally with 1 mL of absolute ethanol and fixed for 24 h. Lungs were subsequently embedded in paraffin blocks and cut into 4 µm sections for hematoxylin and eosin (H&E) staining and immunohistochemistry (IHC).

### 4.5. Metastasis Evaluation

To assess lung metastasis in mice with or without pulmonary allergy, we performed several sagittal sections of the lungs stained with hematoxylin and eosin (H&E). Slides were digitally analyzed using Aperio CS (San Diego, CA, USA) digital pathology equipment. Tumor-infiltrated areas (µm^2^) were measured by digital pathology software. This allowed precise quantification and comparison of lung metastatic lesions between allergic and non-allergic groups.

From each hematoxylin and eosin (H&E) stained section, 10 random fields in each metastatic tumor nodule were selected at 40× magnification, and mitoses were counted. Mitotic index was calculated as follows: Mitotic index = Number of mitoses/10.

### 4.6. Periodic Acid–Schiff (PAS) Staining

Formalin-fixed, paraffin-embedded lung tissue sections (5 µm) were deparaffinized, rehydrated, and incubated with periodic acid solution for 5 min, followed by Schiff reagent for 15 min at room temperature. Sections were then counterstained with hematoxylin, dehydrated, and mounted. PAS staining highlights mucopolysaccharides, including mucus, in magenta, allowing the identification of goblet cells in the airway epithelium.

### 4.7. Immunohistochemistry

Immunohistochemical (IHC) staining was performed manually. Lung sections were baked at 60 °C for 20 min, deparaffinized in xylene and rehydrated through a graded alcohol series. Antigen retrieval was performed by heating tissue sections in citrate buffer pH 6.0 at 95 °C. Endogenous peroxidases were blocked in 3% H_2_O_2_, while non-specific binding was blocked with 2% horse serum for 60 min with shaking at room temperature in a humid chamber. Sections were incubated overnight at 4 °C with anti-TGF-β (ab92486, Abcam, Cambridge, UK) or anti-PCNA antibodies, recognizing the most conserved isoform (ab2426, Abcam, Cambridge, UK). Subsequent washes were performed with 1× PBS and secondary antibody from Vector Laboratories (Burlingame, CA, USA) was incubated for 30 min. Then, sections were revealed with an ImmPRESS HRP Horse Anti-Rabbit IgG Peroxidase Polymer Detection Kit from Vector Laboratories and the DAB Detection System (Vector Laboratories), counterstained with hematoxylin, dehydrated through an alcohol series, and mounted.

### 4.8. Digital Pathology Analysis and Automated Image Quantification

Slides were scanned at 40× in an Aperio ScanScope CS2 whole-slide scanner (Leica Biosystems, Buffalo Grove, IL, USA). Tumor areas were delineated by two expert pathologists. Slides were analyzed in Aperio ImageScope visualization software (version 12.3, Leica Biosystems). Annotations were processed following the Aperio nuclear algorithm (version 9.2, Leica Biosystems), with default settings (“Nuclear Algorithm, User’s Guide” Leica Biosystems, MAN-0338, Revision 8; 5 August 2015). Staining intensity and the percentage of labeled targets were calculated by digitally analyzing color intensity. Input parameters were pre-configured for brown color quantification: [Minimum nuclear size (μm^2^) = 10; Maximum nuclear size (μm^2^) = 1000; Minimum roundness = 0.01; Minimum compactness = 0; Minimum elongation = 0.4; Weak threshold (1+) = 220; Moderate threshold (2+) = 210; Strong threshold (3+) = 195]. The algorithm detects nuclear staining and quantifies pixel intensity, classifying them as: weak (yellow), moderate (orange), strong (red), and negative (blue). Results were obtained as staining intensity from 0 to 3 and statistical analyses were performed on mean values. Data are presented as nuclear intensity/µm^2^ [52]. The parameters of the digital pathology algorithm are shown in the Appendix A.

### 4.9. Database Analysis

To investigate the potential roles of TGF-β and PCNA in lung metastasis and overall survival we compared gene expression in the normal and metastatic tissues of 399 samples from lung cancer patients using TNMplot (https://tnmplot.com) [53] (accessed on 1 December 2024). The association between TGF-β and PCNA and prognosis of osteosarcoma patients was analyzed by the Kaplan–Meier plotter (http://kmplot.com/analysis/) [54] (accessed on 1 December 2024). A total of 583 and 714 patients were analyzed for gene expression of TGF-β and PCNA respectively. The low and high expression cohorts contrasted for OS were split by a percentile-based best cutoff search with false discovery rate (FDR) of 1% measured by the Benjamini–Hochberg method for multiple comparisons. Other parameters included biased array exclusion without restrictions on subtypes or treatment. However, in order to account for data heterogeneity, i.e., biological variation, we adjusted survival by multivariate Cox regression analysis, including clinicopathological characteristics such as histology, stage, gender, and smoking history. Biomarker, time and status data were downloaded to construct customized Kaplan–Meier curves and validate results in external software (SPSS version 25 IBM Corp., USA and R language version 4.4.1). Additionally, PCNA gene expression in non-metastatic vs. metastatic settings was analyzed from the GEO database (https://www.ncbi.nlm.nih.gov/geo/) (accessed on 1 December 2024) with accession number GSE42352 whereas OS was contrasted in high vs. low PCNA levels from the GSE21257 dataset. Expression profiles of multiple genes related to PCNA and cell proliferation pathways in these OS patients were downloaded from the GEO database, and a heatmap of differentially expressed genes was constructed using R 4.4.1. Kaplan–Meier curves and Log rank tests were performed and hazard ratios (HR) with 95% confidence intervals (CI) and *p*-values were calculated. Since only individual datasets were used, i.e., not merged, batch effect correction was not required.

### 4.10. Statistical Analysis

Normality was evaluated by Kolmogorov–Smirnov (K-S) analysis as well as histograms, stem-and-leaf, and Q-Q plots before applying parametric or non-parametric tests. Normal data were analyzed with Prisma^®^ 8.0.1 software and expressed as mean ± standard deviation. Statistical comparisons were performed by Student’s *t* test (two groups) or ANOVA (analysis of variance) and Tukey’s test (three or more groups). For non-normal data, the Mann–Whitney U test was performed for two-group contrasts. Bonferroni correction was used for multiple comparisons (two genes) with adjusted α = 0.05/2 = 0.025. Effect sizes were calculated either as Cohen’s d or as Cliff’s delta for parametric and non-parametric data respectively. Each classifier was ranked on a three-level scale as follows for Cohen’s d: small (d = 0.2), medium (d = 0.5), and large (d ≥ 0.8) [55] and Cliff’s delta: small (δ: 0.11–0.27), medium (δ: 0.28–0.42) and large (δ > 0.43) [56].

## 5. Conclusions

In conclusion, this study highlights the relevance of considering specific interactions between immune-mediated diseases, such as pulmonary allergic inflammation, and cancer progression. Understanding these interactions in detail may provide valuable insights into potential associations between TGF-β signaling and osteosarcoma outcomes, particularly in patients with concomitant conditions such as pulmonary allergic inflammation. However, additional studies are required to establish the direct roles of TGF-β in pulmonary metastasis.

## Figures and Tables

**Figure 1 ijms-26-05073-f001:**
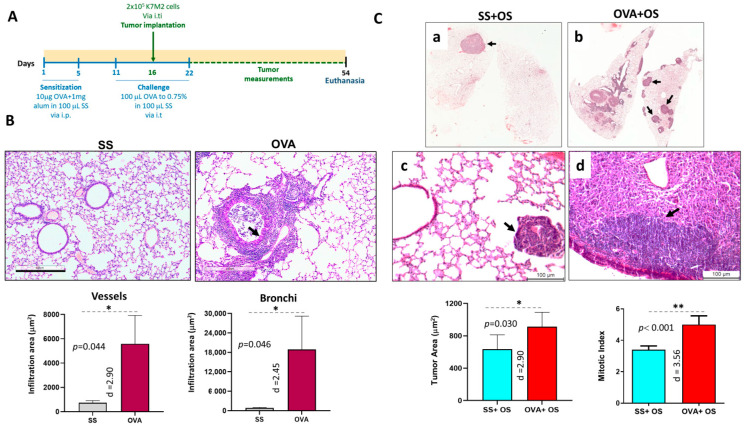
Allergic airway inflammation promotes lung metastasis in the experimental mouse model of osteogenic sarcoma. (**A**) Overview of the experimental design and groups. A murine model of pulmonary allergic inflammation and osteosarcoma (OVA + OS) was established using male BALB/c mice. Mice were sensitized with ovalbumin (OVA) and aluminum hydroxide (Alum) via intraperitoneal (i.p.) injection on days 1 and 5. Allergen challenges were administered intratracheally (i.t.) on day 11 (prior to tumor induction) and day 22 (post-tumor induction). Osteosarcoma was induced on day 16 by intratibial (i.ti.) injection of syngeneic K7M2 osteosarcoma cells. Mice were euthanized at the end of the study via exsanguination. Experimental groups included: SS (saline control), OVA, SS + OS (tumor only), and OVA + OS (allergy + tumor). (**B**) Representative lung histology comparing SS and OVA groups, PAS staining of lung sections shows normal histology in SS controls and extensive peribronchial and perivascular inflammation with mucus production (arrows) in OVA-treated mice. Quantitative histomorphometry confirmed a significantly larger inflammatory area in the OVA group, than in control group (* *p* < 0.05). (**C**) Lung metastasis histology. Representative H&E micrographs of the indicated groups, top figures show low-power micrographs of the lung with one mid-sized metastasis (arrow) in the lung of SS + OS mouse (micrograph a), while numerous variable-sized metastases (arrows) are seen in OVA + OS mouse (micrograph b). In the middle panel are shown higher magnification micrographs, the lung of SS + OS mouse (micrograph c) show small metastatic nodules (arrow) constituted by polyhedral cells with large hyperchromatic nucleus; while the lung of OVA + SS mouse (micrograph d) show a large metastasis (arrow) with bigger cuboidal cells near to a bronchus surrounded by inflammatory cells (arrow) and cellular debris in the lumen that correspond to allergic inflammation. Quantitative morphometry (bottom panel) revealed significantly larger metastatic areas and higher mitotic indices in the OVA + OS group compared to SS + OS. Effect sizes (Cohen’s d) are shown beneath each graph (* *p* < 0.05; ** *p* < 0.01).

**Figure 2 ijms-26-05073-f002:**
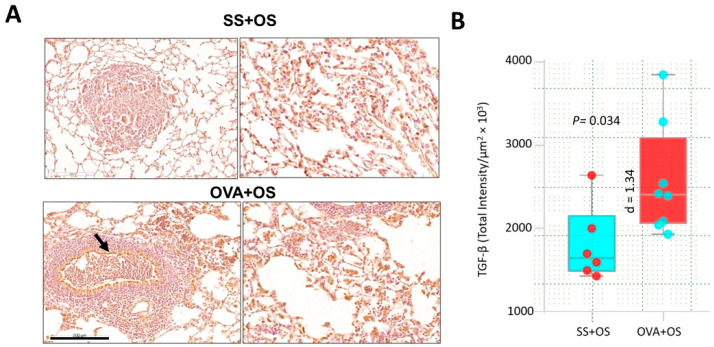
TGF-β overexpression is associated with more lung metastasis from osteosarcoma. (**A**) Representative micrographs of TGF-β detection by immunohistochemistry in the lungs of mice with osteosarcoma and allergy induced by OVA (OVA + OS) compared with mice without allergy (SS + OS). Top left panel shows low-power micrograph from SS + OS mice showing a mid-sized metastatic nodule with some neoplastic positive cells. Top right panel shows high-power micrograph from the same lesion illustrating that the strongest immunostaining is observed in the cytoplasm of large cells. Bottom left panel shows low-power micrograph of OVA + OS mouse. The strongest TGF-β immunostaining is seen in the bronchial epithelium (arrow) and metastatic nodule. Bottom right panel shows high-power micrograph from the same OVA + OS mouse, all neoplastic cells show strong immunostaining, as well as the bronchial epithelium and blood vessel layer. (**B**) Digital pathology-confirmed TGF-β upregulated expression in lungs of OVA + OS vs. SS + OS mice. The difference was statistically significant with a large effect size presented as Cohen’s d.

**Figure 3 ijms-26-05073-f003:**
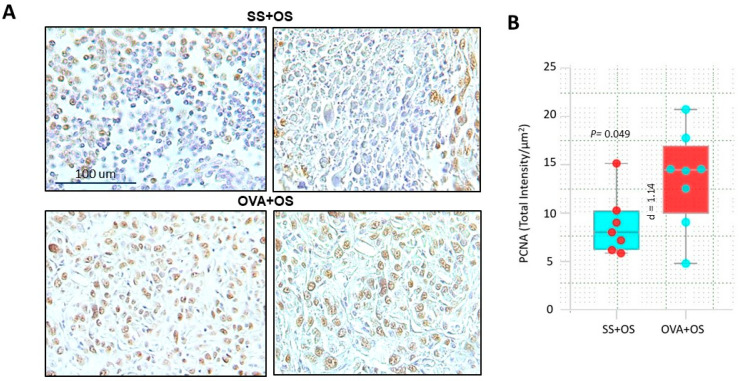
Proliferative cell marker PCNA overexpression correlates with lung metastasis during allergic lung inflammation. (**A**) Low and high-power representative micrographs of PCNA detection by immunohistochemistry in metastatic pulmonary nodules from the indicated mice. Some neoplastic cells show positive nuclear immunostaining in SS + OS mice, while many cells are positive in OVA + OS mice. (**B**) Digital pathology-confirmed higher count of PCNA-positive cells in metastatic nodules from OVA + OS mice vs. SS + OS mice. The difference was statistically significant with large effect size presented as Cohen’s d.

**Figure 4 ijms-26-05073-f004:**
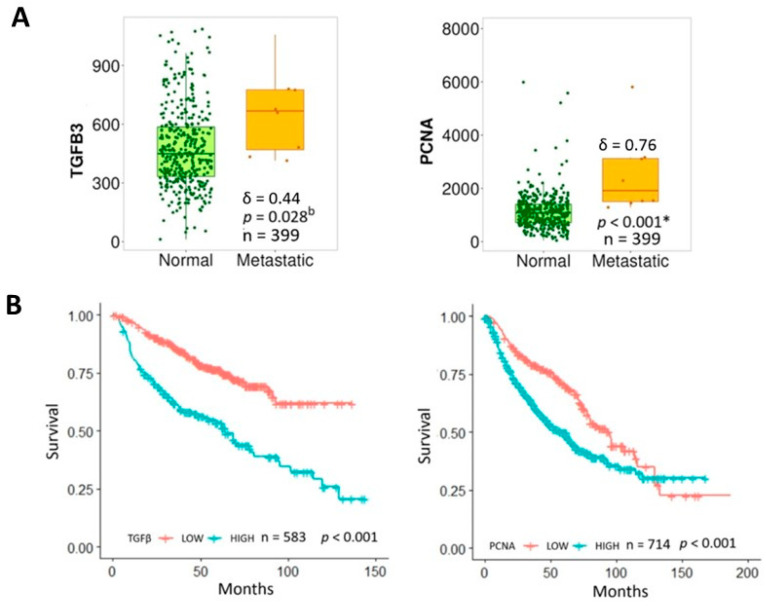
Expression and prognostic value of TGFβ and PCNA in lung metastasis. (**A**) Differential gene expression of TGFβ and PCNA mRNA in normal vs. metastatic lungs from patients. Gene chip data from TNMplot web tool (https://tnmplot.com) (accessed on 1 December 2024). Comparisons were performed by Mann Whitney U test. Results were statistically significant with large effect size. (**B**) Overall survival of lung cancer patients by PCNA and TGFβ level based on KM plotter (https://kmplot.com) (accessed on 1 December 2024). (b: borderline significant and *: significant after Bonferroni correction (adjusted α = 0.05/2 = 0.025). Since data distribution is non-parametric, effect sizes are presented as Cliff’s delta (δ).

**Figure 5 ijms-26-05073-f005:**
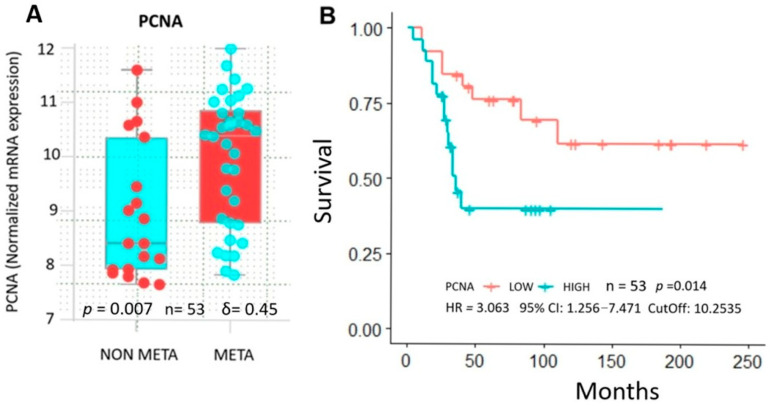
PCNA expression and prognostic impact on patients with osteosarcoma metastasis. (**A**) Higher PCNA expression (mRNA) in patients with metastatic vs. non-metastatic osteosarcoma by Mann–Whitney U test. Results were statistically significant with large effect size (**B**) Overall survival of patients with osteosarcoma stratified by PCNA level through Kaplan– curves and Log rank tests. Data were retrieved from Gene Expression Omnibus (GEO) Database (https://www.ncbi.nlm.nih.gov/geo/, accessions GSE42352 and GSE21257) (accessed on 1 December 2024). Since data distribution is non-parametric, effect size is presented as Cliff’s delta (δ).

**Figure 6 ijms-26-05073-f006:**
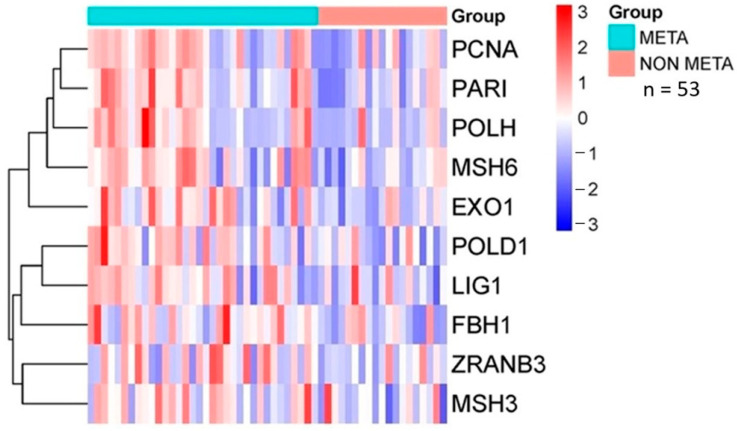
Heatmap of differentially expressed genes directly associated with PCNA and cell proliferation pathways. Clinical samples included metastatic vs. non-metastatic osteosarcoma. Gene expression level is color-coded in red and blue for up- and downregulation, and white indicates no changes. Data were retrieved from pre-chemotherapy biopsies of patients with high-grade osteosarcoma from GEO database accession GSE21257.

## Data Availability

Data are contained within the article.

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
