# Peer review of "TGF-β Induced by Allergic Lung Inflammation Enhances Os-Teosarcoma Lung Metastasis in a Mouse Comorbidity Model"

_ijms, 2025, doi:10.3390/ijms26115073_

Round 1
Reviewer 1 Report
Comments and Suggestions for Authors
Major Comments
- Methodology
- Though the experimental design is mostly good, more explanation on power analysis and sample size justification is required to ensure statistical strength.
- A more clear definition of the selection criteria for BALB/c mice and a justification for their use are needed.
- The methodology part should specify how repeatable the model of allergen-induced lung inflammation is. Did the mice have consistent allergic responses?
- Specifications of the digital pathology algorithm parameters have to be supplied to enable replication. Were they optimized or were the default settings used?
- More details on probe choice for PCNA expression study are required to verify that only relevant isoforms were investigated.
- Data Analysis
- The paper should clearly state if normality tests were run before the use of parametric or non-parametric tests.
- Especially in genomic research, there is insufficient information about the use of multiple comparison corrections (e.g., Bonferroni or FDR adjustments).
- Though the authors claim significant differences among groups, effect sizes must be included to more accurately evaluate the biological significance.
- The paper includes bioinformatics studies based on KM plotter and TNMplot. Still, there is no mention of dataset heterogeneity or batch effect control.
- Results
- The manuscript effectively establishes a link among allergic lung inflammation, TGF-β overexpression, and osteosarcoma metastasis. The role of TGF-β in this pathway remains conjectural. Consider include functional validation tests with TGF-β inhibitors or knockout animals.
- The assertion that TGF-β is a "potential therapeutic target" deserves further validation. Have any in vivo investigations focusing on TGF-β been performed?
- The study fails to address significant confounders, like systemic inflammation and immunological suppression, which may possibly lead to metastasis.
- Figures 4 and 5 depict gene expression and survival outcomes; however, the sample sizes of each dataset must be explicitly stated.
- The study provides significant insights into the relationship between inflammation and metastasis; however, it might be enhanced by comparisons with other malignancies in which TGF-β is implicated.
- Though they appear to conflict with studies suggesting an anti-tumor role of allergic inflammation, the findings are in line with previous TGF-β research in the tumor microenvironment. The discussion should address this difference.
Minor Comments
- Figures should exhibit uniform formatting, encompassing consistent labeling of p-values and statistical significance.
- The reference section has a variety of citation styles, with some entries using whole journal titles while others are shortened. Maintain adherence to journal specifications.
- Abbreviations like PCNA and TGF-β should be defined at their first mention.
- Figures 1 and 2 present significant histological images; nevertheless, they should have scale bars for reference.
- Figure captions should be more comprehensive to ensure the figures are comprehensible without referring to the main text.
- Several tables exhibit ambiguity in their headings, especially those displaying statistical comparisons.
Comments on the Quality of English Language
- Some sentences are excessively intricate and might be made clearer. Example: "These findings highlight potential roles of TGF-β and PCNA as biomarkers in osteosarcoma progression and patient survival." Consider rewording for clarity.
- The manuscript has sporadic grammatical errors (e.g., "high production of TGF is observed in allergic lung inflammation like asthma" should be "high production of TGF-β is observed in allergic lung inflammation, such as asthma").
- There are inconsistencies in verb tenses, particularly when contrasting previously published studies with current findings.
Author Response
Dear Editor and Reviewer,
We appreciate your effort in reviewing our manuscript and providing insightful comments making possible the current version. The authors have considered all comments from reviewers and addressed them carefully. We hope the improved manuscript meets your high standards. Below we provide point-by-point responses while corrections in the manuscript appear highlighted in light blue.
Sincerely,
Dr. Sara Huerta-Yepez
Reviewer 1
Major Comments
Methodology
Comment 1. Though the experimental design is mostly good, more explanation on power analysis and sample size justification is required to ensure statistical strength.
Response 1. Thank you for the comment. We included a paragraph about sample size calculation based on power analysis and resource equation on page 4, Subsection 2.2 Animals and ethical statements
Comment 2. A more clear definition of the selection criteria for BALB/c mice and a justification for their use are needed.
Response 2. We have included a clearer justification for the use of BALB/c mice as a model in the revised version of the manuscript on page 4, Subsection 2.3. Allergic Airway Inflammation Model.
Comment 3. The methodology part should specify how repeatable the model of allergen-induced lung inflammation is. Did the mice have consistent allergic responses?
Response 3. The allergen-induced lung inflammation model is well standardized in our research group, and we have several published articles using the same model. We have added relevant references to support this model on page 4, Subsection 2.3. Allergic Airway Inflammation Model.
Comment 4. Specifications of the digital pathology algorithm parameters have to be supplied to enable replication. Were they optimized or were the default settings used?
Response 4. The parameters of the digital pathology algorithm have been included in a supplementary table 1 for clarity and to allow replication. Page 5, Subsection 2.7. Digital Pathology Analysis and Automated Image Quantification.
Comment 5. More details on probe choice for PCNA expression study are required to verify that only relevant isoforms were investigated.
Response 5. We have now included more details on the antibody selection for the PCNA expression study in the revised manuscript. Specifically, we used a probe that targets the most relevant isoforms of PCNA, ensuring that only the isoforms associated with the studied condition were investigated. Page 5, Subsection 2.7. Immunohistochemistry.
Data Analysis
Comment 1. The paper should clearly state if normality tests were run before the use of parametric or non-parametric tests.
Response 1. We provided an explanation of how normality was evaluated by multiple methods, on page 6 Subsection 2.9 Statistical analysis.
Comment 2. Especially in genomic research, there is insufficient information about the use of multiple comparison corrections (e.g., Bonferroni or FDR adjustments).
Response 2. We included a Bonferroni correction as described on page 6 Subsection 2.9 Statistical analysis and Figure 4 on page 10.
Comment 3. Though the authors claim significant differences among groups, effect sizes must be included to more accurately evaluate the biological significance.
Response 3. Effect sizes were measured and classified. We describe this on page 6 Subsection 2.9 Statistical analysis and they were included in every figure.
Comment 4. The paper includes bioinformatics studies based on KM plotter and TNMplot. Still, there is no mention of dataset heterogeneity or batch effect control.
Response 4. Since datasets used in this study were not merged; batch effect correction was not required as stated on page 6, Subsection 2.8 Database Analysis. Additionally, in order to account for biological heterogeneity, we analyzed overall survival through multivariate Cox regression adjusting for several potential confounding factors as described on page 5 of Subsection 2.8 Database Analysis.
Results
Comment 1. The manuscript effectively establishes a link among allergic lung inflammation, TGF-β overexpression, and osteosarcoma metastasis. The role of TGF-β in this pathway remains conjectural. Consider include functional validation tests with TGF-β inhibitors or knockout animals.
Response 1. We appreciate the reviewer suggestion. A second study is planned in which we will use specific TGF-β inhibitors to demonstrate the role of TGF-β more definitively in lung metastasis. For this reason, we have modified the title of the manuscript to better reflect the results obtained in this initial study.
Comment 2. The assertion that TGF-β is a "potential therapeutic target" deserves further validation. Have any in vivo investigations focusing on TGF-β been performed?
Response 2. The reviewer is right. In the updated version of the manuscript, we have revised the sentences to provide more concise and accurate wording that better reflects our findings. As previously mentioned, a second study is planned in which we will use specific TGF-β inhibitors to demonstrate the role of TGF-β more definitively in lung metastasis. Accordingly, we have modified the title of the manuscript to better align with the results obtained in this initial study.
Comment 3. The study fails to address significant confounders, like systemic inflammation and immunological suppression, which may possibly lead to metastasis.
Response 3. The reviewer is correct; in this initial study, we did not include other factors that may contribute to metastasis, such as systemic inflammation and immunosuppression. In a follow-up study, we are planning to specifically block TGF-β in order to more definitively establish its role in lung metastasis. We have now addressed these potential confounding variables in the discussion section to provide greater clarity. Page 14, Discussion Section.
Comment 4. Figures 4 and 5 depict gene expression and survival outcomes; however, the sample sizes of each dataset must be explicitly stated.
Response 4. We have included samples size for each dataset accordingly.
Comment 5. The study provides significant insights into the relationship between inflammation and metastasis; however, it might be enhanced by comparisons with other malignancies in which TGF-β is implicated.
Response 5. We have included a paragraph addressing TGF-β´s role in stimulating metastasis in multiple malignancies on the Discussion Section on page 14.
Comment 6. Though they appear to conflict with studies suggesting an anti-tumor role of allergic inflammation, the findings are in line with previous TGF-β research in the tumor microenvironment. The discussion should address this difference.
Response 6. Although our findings may appear to conflict with studies suggesting an anti-tumor role for allergic inflammation, they are consistent with previous research on the pro-metastatic role of TGF-β in the tumor microenvironment. We now address this apparent discrepancy in the discussion section to provide a balanced interpretation of the results within the broader context of existing literature. Page 13, Discussion Section.

Reviewer 2 Report
Comments and Suggestions for Authors
The authors report that allergic airway inflammation increased lung metastasis of osteosarcoma implants, which may be related to increased TGF-β-production during allergic inflammation in a OVA mouse model.
There are major issues, which need to be addressed:
- Abstract: Please reduce the text and add a graphical abstract to underline the experimental and human data base approach and main findings.
- The description of the OVA allergy mouse model is not properly documented, especially the cell recruitment, cytokine expression and T2 and T17 differentiation is not considered. This data are crucial to draw the conclusion that allergic inflammation is involved in the spread of osteosarcoma implants.
- The role TGF-β-production is based only on IHC staining, which not sufficient. At least, protein and gene expression data are necessary to document the role of TGF-β-production.
- Figure 1 A: The resolution of histology is not sufficient and there are not sufficient data to characterize the allergic mouse model, this is mandatory.
- Figure 1 B: The resolution of histology is not sufficient and I cannot affirm the conclusions.
- Overall, the conclusion of TGF-β-dependent tumor growth is not supported, and it is only an association study.
- The data base analysis are not convincing, but I am not an expert.
Author Response
Dear Editor and Reviewer,
We appreciate your effort in reviewing our manuscript and providing insightful comments making possible the current version. The authors have considered all comments from reviewers and addressed them carefully. We hope the improved manuscript meets your high standards. Below we provide point-by-point responses while corrections in the manuscript appear highlighted in light blue.
Sincerely,
Dr. Sara Huerta-Yepez
Reviewer 2
Minor Comments
Comment 1. Figures should exhibit uniform formatting, encompassing consistent labeling of p-values and statistical significance.
Response 1. We have added p-value as well as sample and effect size on every figure as suggested.
Comment 2. The reference section has a variety of citation styles, with some entries using whole journal titles while others are shortened. Maintain adherence to journal specifications.
Response 2. We have now revised the reference section to ensure consistency in citation style, following the journal’s formatting guidelines throughout.
Comment 3. Abbreviations like PCNA and TGF-β should be defined at their first mention.
Response 3. We have now defined all abbreviations, including PCNA and TGF-β, at their first mention in the manuscript. Pages 2 and 3, Introduction Section.
Comment 4. Figures 1 and 2 present significant histological images; nevertheless, they should have scale bars for reference.
Response 4. We have now included scale bars in Figures 1 and 2 to provide proper reference for the histological images.
Comment 5. Figure captions should be more comprehensive to ensure the figures are comprehensible without referring to the main text.
Response 5. We have revised the figure captions to make them more comprehensive, ensuring that the figures are understandable without the need to refer to the main text.
Comment 6. Several tables exhibit ambiguity in their headings, especially those displaying statistical comparisons.
Response 6. We have highlighted statistically significant results as well as effect sizes in all headings.
Comments on the Quality of English Language
Comment 1. Some sentences are excessively intricate and might be made clearer. Example: "These findings highlight potential roles of TGF-β and PCNA as biomarkers in osteosarcoma progression and patient survival." Consider rewording for clarity.
Response 1. As suggested by the reviewers, we have revised the original sentences to reduce complexity and improve clarity. In the updated version of the manuscript, these sentences have been replaced with more concise and accurate wording that better reflects our results.
Comment 2. The manuscript has sporadic grammatical errors (e.g., "high production of TGF is observed in allergic lung inflammation like asthma" should be "high production of TGF-β is observed in allergic lung inflammation, such as asthma").
Response 2. We have carefully revised the manuscript to correct grammatical errors, including the example mentioned, improving overall clarity and accuracy throughout the text.
Comment 3. There are inconsistencies in verb tenses, particularly when contrasting previously published studies with current findings.
Response 3. We have reviewed and corrected verb tenses throughout the manuscript to ensure consistency, particularly when contrasting previously published studies with our current findings.
Comments and Suggestions for Authors
The authors report that allergic airway inflammation increased lung metastasis of osteosarcoma implants, which may be related to increased TGF-β-production during allergic inflammation in a OVA mouse model.
There are major issues, which need to be addressed:
Comment 1. Abstract: Please reduce the text and add a graphical abstract to underline the experimental and human data base approach and main findings.
Response 1. We have shortened the abstract to make it more concise and have added a graphical abstract to highlight the experimental and human data-based approach, as well as the main findings.
Comment 2. The description of the OVA allergy mouse model is not properly documented, especially the cell recruitment, cytokine expression and T2 and T17 differentiation is not considered. This data are crucial to draw the conclusion that allergic inflammation is involved in the spread of osteosarcoma implants.
Response 2. We have provided a more detailed description of the OVA allergy mouse model that our group has used in previous publications. We have also included the relevant references to support this. Page 4, Subsection 2.3. Allergic Airway Inflammation Model.
Comment 3. The role TGF-β-production is based only on IHC staining, which not sufficient. At least, protein and gene expression data are necessary to document the role of TGF-β-production.
Response 3. Given that the role of TGF-β production is based solely on IHC staining and gene expression from public data bases, we can only discuss the association of TGF-β with lung metastasis. For this reason, we have changed the title of the manuscript to better reflect this association.
Comment 4. Figure 1 A: The resolution of histology is not sufficient and there are not sufficient data to characterize the allergic mouse model, this is mandatory.
Response 4. We have improved resolution of histological images in Figure 1A, providing additional data to better characterize the allergic mouse model.
Comment 5. Figure 1 B: The resolution of histology is not sufficient, and I cannot affirm the conclusions.
Response 5. We have improved resolution of histological images in Figure 1B to ensure clarity and support the conclusions more effectively.
Comment 6. Overall, the conclusion of TGF-β-dependent tumor growth is not supported, and it is only an association study.
Response 6. The reviewer is correct, and we acknowledge our study presents association rather than direct evidence of TGF-β-dependent tumor progression. For this reason, we have modified the title of the manuscript to better align with the results presented.
Comment 7. The data base analysis are not convincing, but I am not an expert.
Response 7. We have improved analysis and descriptions accordingly. Figures 4,5 and 6 on pages 10-12, Subsection 3.4 database analysis.

Round 2
Reviewer 1 Report
Comments and Suggestions for Authors
None
Author Response
Dear Editor and Reviewer,
We appreciate your effort in reviewing our manuscript and providing insightful comments making possible the current version. The authors have considered all comments from reviewers and addressed them carefully. We hope the improved manuscript meets your high standards. Below we provide point-by-point responses while corrections in the manuscript appear highlighted in light blue.
Sincerely,
Dr. Sara Huerta-Yepez
Reviewer 1. No comments
Thank you.

Reviewer 2 Report
Comments and Suggestions for Authors
The authors replied to the questions; The extensive revision improved the quality of the manuscript. The Fig 1 however should edited with higher resolution in the final submission;
Author Response
Dear Editor and Reviewer,
We appreciate your effort in reviewing our manuscript and providing insightful comments making possible the current version. The authors have considered all comments from reviewers and addressed them carefully. We hope the improved manuscript meets your high standards. Below we provide point-by-point responses while corrections in the manuscript appear highlighted in light blue.
Sincerely,
Dr. Sara Huerta-Yepez
Reviewer 2.
Comment 1. The authors replied to the questions; The extensive revision improved the quality of the manuscript. The Fig 1 however should edited with higher resolution in the final submission.
Response 1. We thank the reviewer for the positive feedback and helpful suggestions. In response to the comment regarding Figure 1, we have now replaced it with a higher-resolution version in the revised manuscript to ensure clarity and improved quality for final submission.
